# The Early Detection of Malignant Transformation of Potentially Malignant Disorders: Oral Lichen Planus

**DOI:** 10.3390/cancers17091489

**Published:** 2025-04-28

**Authors:** Camilla Lüdecke, Heinrich Neumann, Torsten W. Remmerbach

**Affiliations:** 1Department of Oral and Maxillofacial and Facial Plastic Surgery, Section of Clinical and Experimental Oral Medicine, University Hospital Leipzig, Liebigstraße 10-14, 04103 Leipzig, Germany; camilla.luedecke@medizin.uni-leipzig.de; 2Institute of Pathology, Cytopathology and Molecular Diagnostics, Merzenicher Straße 37, 52351 Düren, Germany; heinrich.h.neumann@gmail.com

**Keywords:** oral lichen planus, oral potentially malignant lesions, oral squamous cell carcinoma, DNA image cytometry, cytology, early detection

## Abstract

The aim of this study was to evaluate the usefulness of close clinical surveillance intervals combined with oral brush biopsies to enable early detections of malignant transformations in patients with oral lichen planus performed in our clinic. Furthermore, we compared it to recall intervals given in the literature to be able to give a recommendation.

## 1. Introduction

Lichen planus is an idiopathic immunemediated inflammatory disorder with a prevalence of around 1.01% [1] (p. 818). It is a chronic disease with periods of remission and relapse [2] (p. 240) that predominantly affects middle-aged women. The disease may involve the mouth, skin, nails, hair, or genitals. Usually, oral manifestations arise bilaterally on the buccal mucosa. In recent papers, the reticular, plaque, and papular clinical forms are summarized as white type. Most frequently is the reticular, characterized by lacy, thin white lines also known as Wickham striae. Less common are the red types containing atrophic, bullous, and erosive lesions, which are often sensitive or painful. Because oral lichen planus (OLP) is a dynamic disorder where both white (keratotic) and red (atrophic/erosive) areas coexist or may change over time, the categorization of patients on the basis of clinical form is not always easy and feasible [3] (p. 823). Up to this day, OLP is challenging to treat and there are only medications that reduce the symptomatic cases [4] (p. 311) [5] (p. 103). Recent Cochrane reviews found a lack of strong evidence supporting the effectiveness of any treatment, even for symptomatic OLP [6] (p. 14) [7] (p. 13). Hence, the reticular form of lichen planus, when asymptomatic, does not require treatment [8] (p. 96).

OLP belongs to the category of oral potentially malignant disorders (OPMDs), albeit with a low risk of malignant transformation according to WHO [9] (p. 1869). It makes up the second largest group of OPMDs after Leukoplakia [10] (p. 11). While few authors still question its malignant potential [11], a transformation rate of around 1–2% is widely accepted among experts [12,13,14,15] (p. 1932) [16] (p. 1911) [17]. Although still controversial, numerous authors have found the atrophic-erosive forms to be predisposed to cancer development [18] (p. 696).

Oral squamous cell carcinomas (OSCCs) make up 90% of all head and neck tumours worldwide. Up until today, the survival rates for oral cancer have been poor and have not improved much despite advances in therapy intervention. At an early stage of diagnosis, the 5-year survival rate for an OSCC is 86%; however, only 30% of oral cancers are diagnosed at this point. Around 52% of cases are not recognized until the tumour is in an advanced stage and has infiltrated the locoregional lymph nodes; in this case, the 5-year survival rate drops to 69%. When the cancer has spread throughout the body, the 5-year survival rate declines even further to 40% [19] (p. 31). Disease-free survival and overall survival rates drop by 50% when comparing T1 to T2 and the above tumours [20] (p. 369). Hence, a key factor in rates not having improved has been the lack of early detections and the consequent failure of valid early treatments for oral cancer.

In 2002, Ulf Mattson et al. stated that, “on a practical and economic basis, a continuous recall of all OLP patients in specialist clinics cannot be justified” [21]. But during the last 20 years of not having a curative treatment [5] (p. 103), alongside the constant risk of malignant transformations, many authors have started supporting the need for lifelong follow-up. It is one of the major objectives of the WHO to lighten the burden of OSCC globally through interventions that can help prevent or detect the disease at an early stage [22] (e7). To attain this WHO objective, routine screening for OPMDs and OC needs to be carried out [23]. Advice concerning the frequency of follow-ups given in the literature vary largely from 1 to 6 times annually [24,25] (p. 578) [26] (p. 547).

The aim of this study was to evaluate the usefulness of the surveillance program performed by our clinic. We propose close follow-ups for patients affected by oral lichen planus combined with regular biopsy, and, in doubtful cases, we propose oral brush-based cytology to enable earlier detections of transformations to OSCCs compared to the literature. This may lead to reduced mortality and morbidity and a better survival rate in our clinic compared to the literature.

## 2. Materials and Methods

A total of 414 patients with a clinical diagnosis of oral lichen planus that visited the University Hospital of Leipzig between 1993–2022 were enrolled into this retrospective study. After applying our criteria for inclusion, the records of patients who did not meet our criteria were excluded from this study, leaving us with a cohort of 297 patients.

### 2.1. Diagnostic Criteria for Inclusion

1. A diagnosis of OLP based on the following criteria: Each patient was examined visually by an oral surgeon, applying strict clinical and histopathological diagnostic modified WHO criteria [27] (p. 510) [28] (p. 348). Oral lichenoid lesions were ruled out by the removal of potential triggers (e.g., Amalgam) or were excluded from this study. In cases of missing histopathological findings, a scalpel biopsy was performed to confirm the diagnosis of oral lichen planus. Moreover, a brush biopsy was performed to control the biological behaviour of cells.

The cytological specimens were classified as “positive”, “suspicious”, “doubtful”, or “negative” by experienced cytopathologists. If the cytology results stated “suspicious”, “doubtful”, or “positive” (83 patients, *n* = 132), a further examination using DNA image cytometry was undertaken (at the Department of Cytopathology at the University Hospital of Düsseldorf) to rule out the existence of cells with an abnormal ploidy status. The slides were Feulgen stained, 30–40 normal reference lymph or epithelial cells were used to determine the mean DNA content of 2c. About 300 cells were then analysed and their DNA content measured. The criteria for aneuploidy were a stem line with more than 10% deviation of a diploid (2c) or tetraploid (4c) cells or a single cell containing DNA > 9c.

For further details of the clinical and pathological procedures, see Bechstedt et al. (2022) [29].

2. The follow-up period lasted > 6 months from the clinical and histopathological diagnosis.

3. The absence of any kind of head or neck cancer in the patient’s medical history.

### 2.2. Data Collection

All patients were inspected at the outpatient clinic of our department by an experienced oral surgeon following standardized procedures. The patients had regular follow-up examination every 6 months. Furthermore, the patients attended a check-up with their dentist every 6 months. Overall, they were monitored about every 3 months. Biopsies were taken to enable a primary histopathological diagnosis. Additionally, brush biopsies were taken during follow-up from clinically suspicious regions to enable a non-invasive, effective way of detecting a chance in ploidy status. The results were compared to the results from the histopathological examination of the respective area.

For each patient, the following data were recorded: their age at the time of diagnosis, gender; smoking status; systemic chronic disease status; former cancer diagnosis status; medication usage; use of drugs status; the clinical aspect of the lesion (clinical form, localization, symptoms, treatment provided, and extraoral manifestation); and the malignant transformation of the lesion.

### 2.3. Data Analysis

Data were entered into an Excel database. Every OSCC that was diagnosed in this study was double-checked retrospectively regarding a former histopathologically proven diagnosis of OLP and OSCC arising at the same site as the OLP lesions beforehand.

## 3. Results

A total of 297 patients were included in this study. A total of 894 brush biopsies, and an additional 124 tissue biopsies and 130 examinations using DNA image cytometry, were undertaken during follow-up periods. The mean follow-up period was 6.02 years (with periods ranging from 6 months up to 22.2 years). The median follow-up time was 11.4 years. A total of 226 patients were female, and only 71 were male. Overall, 46 (15.5%) patients had extraoral manifestations of lichen planus. Further data concerning the medical backgrounds of this cohort can be found in Table 1. On average, the patients’ age at first visit was 59.6 years, (19–88). Interestingly, of the 146 patients who had a referral to our clinic by general dentists, only 71 patients were diagnosed with oral lichen planus; 48% of the cases were thus misdiagnosed. A total of 22 (7.4%) patients smoked and 43 (14.5%) stated a weekly consumption of alcohol. A malignant transformation occurred in 4 patients, 3 of which were female and 1 was male (Table 2). This leads to a transformation rate of 1.3%, which is comparable to previous study results. The average time between the first diagnosis and malignant transformation was 8.1 years. The mean age at first diagnosis of OSCC amounted to 74.3 years.

Three of the patients undergoing malignant transformation were formerly diagnosed with lichen erosive and one showed lichen atrophicus. Two were diagnosed as having a carcinoma in situ/SIN III, while one patient had a pT1pNoM0 tumour; the fourth patient showed finally a pT3N0M0 tumour. In three of these cases, clinical suspicion led to brush biopsies; afterwards, DNA image cytometry was undertaken, in which all probes occurred as aneuploid regarding both the stem line and single cell aneuploidy. Of these three patients, none had metastasis, needed a radiotherapy or chemotherapy, or developed a relapse. The removal of the tumour was conducted during a single operation with a short stay in hospital, performed under local anaesthesia. The fourth patient was transferred to the ENT department of the clinic after a positive DNA image cytometry result. A lymphatic node was removed and the primary cancer was not found; a carcinoma of unknown primary (CUP) was therefore diagnosed. The follow-up could not be conducted properly by our department. The patient did not attend the follow-up appointments. The patient appeared in our department two years later, the tumour discovered at this time was in an advanced stage. The patient needed a neck dissection and combined adjuvant radio chemotherapy and they required multiple stays in our hospital ward. At the closure of this study, the patient died after palliative treatment. Nonetheless, at the last follow-up visit in 2024, all other OLP-OSCC patients were still alive.

## 4. Discussion

The average time between the first diagnosis and malignant transformation was 8.1 years, a higher result compared to previous studies [30] (p. 646) [12,24], but similar to other studies [31]. The mean age at first diagnosis of OSCC amounted to 74, this being higher compared to patients without OLP, as has previously been reported in the literature [32] (p. 5). This seems reasonable, since the aetiology is commonly known to be habit induced, such as tobacco consumption, smoking, or alcohol abuse, which starts in early adulthood. On the other hand, oral lichen planus develops later in life and the transformation to OSCC is therefore, later compared to non-OLP OSCC.

In our study, all cases with malignant transformation developed out of red lesions of OLP. These finding support reports of a larger likelihood for red types to transform [33]. Therefore, we agree with the idea of close recall intervals, especially for clinically suspicious lesions suggested by other authors, since most neoplasms/early carcinomas are found in macroscopically visible lesions [3] (p. 822). Only 62 patients in our study showed a change of clinical form during follow-up. This could be due to the start and end of the treatment, the intake of triggers, and the possibility of different clinical forms being present at the same time in one patient. In addition, the overall stability of the OLP lesions observed clinically in the present study seems to exclude the direct correlation between neoplastic events and changes in OLP lesions’ appearance over time [34] (p. 331). However, changes in ploidy status or dysplasia are not visible. Therefore, periodic histologic or cytologic diagnostics are needed to detect these microscopic changes toward malignancy to eliminate guesswork about which lesion requires surgical biopsy and lessen the delay in referring patients [23].

The average non-OLP OSCC is discovered at stages III–IV, which results in treatment complications, a poor prognosis, and financial burden [23]. Usually, OLP-related OSCC is diagnosed at an earlier stage [35] (p. 1196). In three of the patients who took part in our follow-up program, no secondary tumours, no metastasis, and no lymph node metastasis were found and all were Cis/T1, leading to a better overall survival rate, which confirmed the data of other studies [34] (p. 332) [32] (p. 5) [36]. This may be due to the regular check-ups and long-term follow-ups enabling early discoveries. Nevertheless, it must be mentioned that most patients with OLP might drop out of a lifelong follow-up programme. Furthermore, the better prognosis may be due to inherent biologic features of the tumour itself, since they are often grade I (well-differentiated OSCCs) compared to conventional OSCCs, which present typically as grade II (moderately well differentiated OSCCs) [36] (p. 11).

Unfortunately, we were not able to state a 5-year survival rate, because the follow-up interval was not reached. But at the end of this study, all of the patients were still alive, without any tumour relapse, something which has been reported in other studies [34] (p. 331). What can be stated is that the invasiveness of the treatment was very low compared to non-OLP-OSCC, since it is diagnosed at a later stage, reducing costs [37] (p. 348) and increasing saving capacities for healthcare systems, since follow-up visits can be managed by general dentists. Consistent with the literature, our study shows oral brush biopsies are an easy, inexpensive, reliable, and effective tool for detecting OSCCs in their early stages [38] (p. 6658) [29]. Of 297 patients, only 24% (n = 71) were referred with a proper diagnosis, which might indicate the need for further training in diagnosing OPMDs and surveillance options for general dentists.

Although they are consistent with other reports on this topic, several limitations should be considered when interpreting our results. Firstly, this study had retrospective design and lacked a control group with a larger recall interval or the absence a follow-up regime. We were not able to control whether our patients followed our recommendation to visit their dentist every 6 months in addition to the visits they made to our clinic. Furthermore, the study was conducted within a geographically limited area (Germany). Even if our rate of malignant transformations matches that of other studies [39] (ot6), it is likely to be overestimated. Often, only severe and painful cases are referred to our clinic, and with a prevalence of 1% OLP patients in the general population and a transformation rate of 1%, almost all OSCCs would develop from OLP lesions [12], which is simply not the case. Additionally, we had four cases of malignant transformation only, which is a very limited cohort and this needs to be recognized.

Concerning the intervals of recall, the European Association of Oral Medicine suggests regular check-ups [40] (p. 4) and the American Association of Oral Medicine wants patients to be periodically monitored [41]. A more precise statement can be found in a guideline concerning diagnostic and management of OPMDs published by the Association of the Scientific Medical Societies in Germany (AWMF). In terms of the length of time between follow-ups, no longer than 4 months apart is advised [42]. This fits our regime and seems to be an appropriate and useful interval.

## 5. Conclusions

Our results have confirmed the importance of a close surveillance programme for all patients suffering on oral lichen planus, consisting of check-ups every 3 months. The clinical investigation should include brush biopsy in doubtful cases and, if cytologically conspicuous, the additional application of DNA image cytometry. These three components are able to detect malignant transformation in its early intraepithelial and microinvasive phases in the majority of cases, which are in general characterized by a very good prognosis, matching previous studies [3] (p. 822). Therefore, we recommend a close follow-up regime every 3 months, which may be managed by general practitioners performing a routine mucosal screening and brush biopsy within the annual check-up. Especially suspicious lesions should be closely monitored.

## Figures and Tables

**Table 1 cancers-17-01489-t001:** Medical background of the cohort.

Variable	Number (n = 297)	In %
Pre-existing conditions		
Liver	9	3
Rheumatic	31	10.4
DM	41	13.8
Thyroid	77	25.9
Hypertension	150	50.5
Neurologic	33	11.1
Previous tumours	16	5.4
Complete remission of OLP	4	1.3
Smoking status	22	7.4
Alcohol consumption	43	14.5
Extraoral manifestation	46	15.5

**Table 2 cancers-17-01489-t002:** Histopathological and medical background of OLP cases with malignant transformation to OSCC.

Patient	1	2	3	4
Gender	W	M	W	W
Age at diagnosis of OLP	75	72	70	55
Type of OLP	atrophic	erosive	erosive	erosive
Site of OLP	lower lip, vestibule	alveolar ridge	tongue	alveolar ridge
Extraoral OLP sites	no	no	no	no
OLP therapy	none	topical	topical, intralesional, amalgam removal	topical, intralesional
Symptoms	yes	no	yes	yes
Primary site of OSCC	vestibule	alveolar mucosa	tongue	alveolar mucosa
TNM classification	pT1pNoM0	pT1cN0cM0	pTis	2018: CUP cT1N1M0 2021: pT3pN1M0
Follow-up before transformation (yrs)	4	1	2	11
Drug use	none	none	none	alcohol
Systemic diseases	DM, depression, cirrhosis, 1M prolia: osteoporosis	malignant melanoma, cirrhosis, nodular goitre	DM, Hashimoto, hypertonia	DM, depression
Brush biopsy/Cytology	/	doubtful	doubtful	doubtful
DNA ICM aneuploidy	*/*	single cell 9c and stemline	single cell 9c and stemline	2017: stemline 2021: single cell 9c
Histology	*OSCC*	OSCC	OSCC	OSCC

## Data Availability

Data is unavailable due to privacy restrictions.

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
