# Peer review of "The Early Detection of Malignant Transformation of Potentially Malignant Disorders: Oral Lichen Planus"

_cancers, 2025, doi:10.3390/cancers17091489_

Round 1

Reviewer 1 Report

Comments and Suggestions for Authors

Introduction

Author:

Lichen is an idiopathic immune-mediated inflammatory disorder with a prevalence of around 1,27% worldwide [1]

Coment: Today we have evidence-based prevalence data derived from a very extensive and widely cited meta-analysis in the international literature. This citation should be replaced by the following:

González-Moles MÁ, Warnakulasuriya S, González-Ruiz I, González-Ruiz L, Ayén Á, Lenouvel D, Ruiz-Ávila I, Ramos-García P. Worldwide prevalence of oral lichen planus: A systematic review and meta-analysis. Oral Dis. 2021 May;27(4):813-828. doi: 10.1111/odi.13323. Epub 2020 Apr 2. PMID: 32144836.

Authors: OLP belongs to the category of oral potentially malignant disorders (OPMD) albeit with a low risk of malignant transformation according to WHO.

Comments: A WHO consensus meeting was held in 2020 that provides the basis for the current consideration of PLO as an OPMD. This consensus is having a very wide international impact. It should be cited here:

Warnakulasuriya S, Kujan O, Aguirre-Urizar JM, Bagan JV, González-Moles MÁ, Kerr AR, Lodi G, Mello FW, Monteiro L, Ogden GR, Sloan P, Johnson NW. Oral potentially malignant disorders: A consensus report from an international seminar on nomenclature and classification, convened by the WHO Collaborating Centre for Oral Cancer. Oral Dis. 2021 Nov;27(8):1862-1880. doi: 10.1111/odi.13704. Epub 2020 Nov 26. PMID: 33128420.

Authors: While few authors still question its malignant potential [10], a transformation rate of around 1 % is widely accepted among experts. [11] [12] .

Comments: There are now meta-analyses that provide evidence-based information on the malignancy rate of PLO. One of these has been published in the journal Cancers. These meta-analyses should be cited:

González-Moles MÁ, Ramos-García P. An Evidence-Based Update on the Potential for Malignancy of Oral Lichen Planus and Related Conditions: A Systematic Review and Meta-Analysis. Cancers (Basel). 2024 Jan 31;16(3):608. doi: 10.3390/cancers16030608. PMID: 38339358; PMCID: PMC10854587.

Ramos-García P, González-Moles MÁ, Warnakulasuriya S. Oral cancer development in lichen planus and related conditions-3.0 evidence level: A systematic review of systematic reviews. Oral Dis. 2021 Nov;27(8):1919-1935. doi: 10.1111/odi.13812. Epub 2021 Mar 11. PMID: 33616234.

González-Moles MÁ, Ramos-García P, Warnakulasuriya S. An appraisal of highest quality studies reporting malignant transformation of oral lichen planus based on a systematic review. Oral Dis. 2021 Nov;27(8):1908-1918. doi: 10.1111/odi.13741. Epub 2020 Dec 24. PMID: 33274561.

González-Moles MÁ, Ruiz-Ávila I, González-Ruiz L, Ayén Á, Gil-Montoya JA, Ramos-García P. Malignant transformation risk of oral lichen planus: A systematic review and comprehensive meta-analysis. Oral Oncol. 2019 Sep;96:121-130. doi: 10.1016/j.oraloncology.2019.07.012. Epub 2019 Jul 22. PMID: 31422203.

Material and methods

Authors: 1. Diagnosis of OLP based on the following criteria: Each patient was examined by  an oral or maxillo-facial surgeon visually, applying strict clinical and histopathological  diagnostic modified WHO criteria [22]

Comments: Citation 22 by authors van der Wall and van der Meij offers diagnostic criteria that are not supported by the WHO. Among these criteria, OLP is considered to have bilateral lesions and, above all, dysplasia is considered to be a diagnostic exclusion criterion for OLP. Consequently, if the authors use the van der Meij criteria, they are excluding the main risk factor for the development of cancer in OLP, which is none other than epithelial dysplasia. This has been clearly demonstrated in the meta-analysis by Gonzalez Moles published in Oral Oncology in 2019. These criteria are today absolutely questioned by the majority of clinicians and researchers who manage OLP and will necessarily greatly decrease the malignancy rate of the lesions in their series.

Results:

Authors: A malignant transformation occurred in 4 patients, 3 female and one male. ( Table 2 ) Which leads to a transformation rate of 1,3%, comparable to previous study results. [24]

Comment: By applying the inclusion criteria used by the authors, OLP cases with epithelial dysplasia have been excluded. I have previously commented that this diagnostic criterion is not evidence-based and there is no logical reason to apply it. If dysplastic cases have been excluded, this malignancy rate of OLP is most likely underestimated. This should be acknowledged in the discussion.

Discussion

Authors:

who took part in our follow up program, no secondary tumours, no metastasis, no lymph node metastasis were found and all were Cis/T1 , leading to a better overall survival confirming the data of other studies. [28] (p 5)

Comment: This systematic review and meta-analysis should be also referenced here.

González-Moles MÁ, Warnakulasuriya S, González-Ruiz I, González-Ruiz L, Ayén Á, Lenouvel D, Ruiz-Ávila I, Ramos-García P. Clinicopathological and prognostic characteristics of oral squamous cell carcinomas arising in patients with oral lichen planus: A systematic review and a comprehensive meta-analysis. Oral Oncol. 2020 Jul;106:104688. doi: 10.1016/j.oraloncology.2020.104688. Epub 2020 Apr 16. PMID: 32305649.

Authors:

This may be due to the regular check-ups and long term follow up enabling  the early discovery. Unfortunately, we were not able to state a 5-year survival rate, be cause the follow up interval was not reached. But at the end of this study all patients were still alive, without any tumour relapse, which were reported in other studies. [25]

Comment:

There is insufficient evidence that follow-up is responsible for diagnosis at earlier stages. It should be borne in mind that most patients with PLO will drop out of a follow-up programme that should last for the patient's lifetime - there is literature on this issue. The meta-analysis I recommended in the previous commentary states that this better prognosis of PLO carcinomas is probably due to inherent features of the tumour itself. In general, these tumours are more often well-differentiated. A comment on these aspects should be made in the discussion.

Authors:

Therefore, general dentist should be advised to undertake a regular brush biopsy and should be trained in the diagnosis and surveillance of OPMDs. As of now there seems to be a lack of knowledge or awareness.

Comments: There is no evidence for this recommendation. When suspicious lesions appear in an OPMD, a scalpel biopsy should be taken to make a confident assessment of whether or not oral cancer is present. This paragraph should be deleted.

Authors:

Concerning the intervals of recall, the European Association of Oral Medicine sug gests regular check-ups [32] (p 4)

Comment:

It is impossible to find such a citation and should therefore be removed or citated in such a way that it can be analysed by readers.

Authors:

and the American Association of Oral Medicine wants patients to be periodically monitored. [33] .A more precise statement can be found in a guideline concerning diagnostic and management of OPMDs published by the Association of the Scientific Medical Societies in Germany (AWMF). Within a follow up no longer than 4 months apart is advised. [34] This fits our regime and seems to be an appropriate and useful interval.

Comment: It is impossible to find the citation of Weber B.

None of the above citations establishes a follow-up protocol. There is no evidence that three months between visits is better than other follow-up periods. When a periodicity of follow-up is advised, it should be taken into account that frequent visits, e.g. every three months, are very likely to lose many patients, which is very serious in OPMD. Authors should abstain from recommending short follow-up periods. They should focus on reliably informing patients that they have OPMD. They should acknowledge in the paper that very close follow-up periods are accompanied by patient drop-outs. In my personal opinion (of course what I am going to say is not evidence-based either) a yearly frequency of visits is more realistic; patients should have a very easy way of contacting their clinicians, for example through telephone numbers that are answered directly by clinicians, so that in case of changes in the lesion patients can make an earlier appointment. I believe that this is associated with a lower rate of patient loss to follow-up, although, again, this comment of mine is not evidence-based either.

Author Response

Introduction

Author:

Lichen is an idiopathic immune-mediated inflammatory disorder with a prevalence of around 1,27% worldwide [1]

Coment: Today we have evidence-based prevalence data derived from a very extensive and widely cited meta-analysis in the international literature. This citation should be replaced by the following:

González-Moles MÁ, Warnakulasuriya S, González-Ruiz I, González-Ruiz L, Ayén Á, Lenouvel D, Ruiz-Ávila I, Ramos-García P. Worldwide prevalence of oral lichen planus: A systematic review and meta-analysis. Oral Dis. 2021 May;27(4):813-828. doi: 10.1111/odi.13323. Epub 2020 Apr 2. PMID: 32144836.

Response: Thank you for the advice to use the newest data. We changed the citation as you recommended. (p.1, line 38: “1.01% [1] (p 818)”)

Authors: OLP belongs to the category of oral potentially malignant disorders (OPMD) albeit with a low risk of malignant transformation according to WHO.

Comments: A WHO consensus meeting was held in 2020 that provides the basis for the current consideration of PLO as an OPMD. This consensus is having a very wide international impact. It should be cited here:

Warnakulasuriya S, Kujan O, Aguirre-Urizar JM, Bagan JV, González-Moles MÁ, Kerr AR, Lodi G, Mello FW, Monteiro L, Ogden GR, Sloan P, Johnson NW. Oral potentially malignant disorders: A consensus report from an international seminar on nomenclature and classification, convened by the WHO Collaborating Centre for Oral Cancer. Oral Dis. 2021 Nov;27(8):1862-1880. doi: 10.1111/odi.13704. Epub 2020 Nov 26. PMID: 33128420.

Response: We agree completely and added the citation as you recommended. ( p.2, line 54: “[9] (p 1869)”)

Authors: While few authors still question its malignant potential [10], a transformation rate of around 1 % is widely accepted among experts. [11] [12] .

Comments: There are now meta-analyses that provide evidence-based information on the malignancy rate of PLO. One of these has been published in the journal Cancers. These meta-analyses should be cited:

González-Moles MÁ, Ramos-García P. An Evidence-Based Update on the Potential for Malignancy of Oral Lichen Planus and Related Conditions: A Systematic Review and Meta-Analysis. Cancers (Basel). 2024 Jan 31;16(3):608. doi: 10.3390/cancers16030608. PMID: 38339358; PMCID: PMC10854587.

Ramos-García P, González-Moles MÁ, Warnakulasuriya S. Oral cancer development in lichen planus and related conditions-3.0 evidence level: A systematic review of systematic reviews. Oral Dis. 2021 Nov;27(8):1919-1935. doi: 10.1111/odi.13812. Epub 2021 Mar 11. PMID: 33616234.

González-Moles MÁ, Ramos-García P, Warnakulasuriya S. An appraisal of highest quality studies reporting malignant transformation of oral lichen planus based on a systematic review. Oral Dis. 2021 Nov;27(8):1908-1918. doi: 10.1111/odi.13741. Epub 2020 Dec 24. PMID: 33274561.

González-Moles MÁ, Ruiz-Ávila I, González-Ruiz L, Ayén Á, Gil-Montoya JA, Ramos-García P. Malignant transformation risk of oral lichen planus: A systematic review and comprehensive meta-analysis. Oral Oncol. 2019 Sep;96:121-130. doi: 10.1016/j.oraloncology.2019.07.012. Epub 2019 Jul 22. PMID: 31422203.

Response: Thank you for the current literature, we added it accordingly to your comment.( p.2,line 56 :”a transformation rate of around 1-2 % is widely accepted among experts. [12] [13] [14] [15] (p 1932) [16] (p 1911) [17].”)

Material and methods

Authors: 1. Diagnosis of OLP based on the following criteria: Each patient was examined by an oral or maxillo-facial surgeon visually, applying strict clinical and histopathological diagnostic modified WHO criteria [22]

Comments: Citation 22 by authors van der Wall and van der Meij offers diagnostic criteria that are not supported by the WHO. Among these criteria, OLP is considered to have bilateral lesions and, above all, dysplasia is considered to be a diagnostic exclusion criterion for OLP. Consequently, if the authors use the van der Meij criteria, they are excluding the main risk factor for the development of cancer in OLP, which is none other than epithelial dysplasia. This has been clearly demonstrated in the meta-analysis by Gonzalez Moles published in Oral Oncology in 2019. These criteria are today absolutely questioned by the majority of clinicians and researchers who manage OLP and will necessarily greatly decrease the malignancy rate of the lesions in their series.

Response: Excuse our error in citation, of course our clinic uses the internationally accepted WHO criteria. ( p.3, line 95: “[28] (p 348)“)

Nonetheless our retrospective study started in 1993 many different diagnostic statements/ classifications have been made over the years. But our procedure stayed constant. All patients were seen by a Professor for Oral medicine for a clinical diagnosis. Additionally, all patients had a biopsy /histological diagnosis stated. During the follow-up (median 11 years) the diagnosis of OLP was checked over and over. None of the patients showed a dysplasia in their first visit. It only occurred during follow-up in rare cases. In those cases, the patients weren’t excluded from the study.

Results:

Authors: A malignant transformation occurred in 4 patients, 3 female and one male. ( Table 2 ) Which leads to a transformation rate of 1,3%, comparable to previous study results. [24]

Comment: By applying the inclusion criteria used by the authors, OLP cases with epithelial dysplasia have been excluded. I have previously commented that this diagnostic criterion is not evidence-based and there is no logical reason to apply it. If dysplastic cases have been excluded, this malignancy rate of OLP is most likely underestimated. This should be acknowledged in the discussion.

Response: As we stated before, none of the patients showed a dysplasia in their first visit and were therefore excluded. It only occured during follow-up in rare cases. If dysplasia was found in the follow-up probe, the patients weren’t excluded from the study.

Discussion

Authors:

who took part in our follow up program, no secondary tumours, no metastasis, no lymph node metastasis were found and all were Cis/T1 , leading to a better overall survival confirming the data of other studies. [28] (p 5)

Comment: This systematic review and meta-analysis should be also referenced here.

González-Moles MÁ, Warnakulasuriya S, González-Ruiz I, González-Ruiz L, Ayén Á, Lenouvel D, Ruiz-Ávila I, Ramos-García P. Clinicopathological and prognostic characteristics of oral squamous cell carcinomas arising in patients with oral lichen planus: A systematic review and a comprehensive meta-analysis. Oral Oncol. 2020 Jul;106:104688. doi: 10.1016/j.oraloncology.2020.104688. Epub 2020 Apr 16. PMID: 32305649.

 Response: Thank you again for providing further literature. We added it to the citation. ( p.5, line 185: “[37].”)

Authors:

This may be due to the regular check-ups and long term follow up enabling  the early discovery. Unfortunately, we were not able to state a 5-year survival rate, be cause the follow up interval was not reached. But at the end of this study all patients were still alive, without any tumour relapse, which were reported in other studies. [25]

Comment:

There is insufficient evidence that follow-up is responsible for diagnosis at earlier stages. It should be borne in mind that most patients with PLO will drop out of a follow-up programme that should last for the patient's lifetime - there is literature on this issue. The meta-analysis I recommended in the previous commentary states that this better prognosis of PLO carcinomas is probably due to inherent features of the tumor itself. In general, these tumors are more often well-differentiated. A comment on these aspects should be made in the discussion.

Response: Thank you for your comment. We added the point to the discussion and used the Meta-Analysis you recommended to quote.

(p., line 199 ff: “Nevertheless, it must be mentioned most patients with OLP might drop out of a lifelong follow-up programme. Furthermore, the better prognosis may be due to inherent biologic features of the tumour itself, since they are often grade I (well-differentiated oSCC) compared to conventional oSCC which present typically in grade II (moderately well differentiated OSCC). [37] (p 11)“ )

Authors:

Therefore, general dentist should be advised to undertake a regular brush biopsy and should be trained in the diagnosis and surveillance of OPMDs. As of now there seems to be a lack of knowledge or awareness.

Comments: There is no evidence for this recommendation. When suspicious lesions appear in an OPMD, a scalpel biopsy should be taken to make a confident assessment of whether or not oral cancer is present. This paragraph should be deleted.

Response: We altered the paragraph to: (p.6, line 211 ff: “Of 297 patients only 24% (n=71) were referred with a proper diagnosis which might indicate the need for further training in diagnosing OPMDs and surveillance options for general dentists.”)

Authors:

Concerning the intervals of recall, the European Association of Oral Medicine suggests regular check-ups [32] (p 4)

Comment:

It is impossible to find such a citation and should therefore be removed or citated in such a way that it can be analysed by readers.

Response: Please excuse this error in our citation system. You can find the recommendations online https://eaom.eu/education/eaom-handbook/oral-lichen-planus/?v=5f02f0889301. In the section “prevention” you can find the “regular checkups” we are referring to. We edited the citation. The eaom is widely accepted and many colleagues trust their statements and follow their recommendation at least in Germany. Therefore, we see the need to include their statement.

(p.6,line 227: “[40]”)

Authors:

and the American Association of Oral Medicine wants patients to be periodically monitored. [33] .A more precise statement can be found in a guideline concerning diagnostic and management of OPMDs published by the Association of the Scientific Medical Societies in Germany (AWMF). Within a follow up no longer than 4 months apart is advised. [34] This fits our regime and seems to be an appropriate and useful interval.

Comment: It is impossible to find the citation of Weber B.

Response: Please excuse the citation error. Again, this is a high standard society in Germany. You can find the guidelines online (https://register.awmf.org/de/leitlinien/detail/007-092) We edited the citation accordingly. (p.6, line 231: “[42]”)

None of the above citations establishes a follow-up protocol. There is no evidence that three months between visits is better than other follow-up periods. When a periodicity of follow-up is advised, it should be taken into account that frequent visits, e.g. every three months, are very likely to lose many patients, which is very serious in OPMD. Authors should abstain from recommending short follow-up periods. They should focus on reliably informing patients that they have OPMD. They should acknowledge in the paper that very close follow-up periods are accompanied by patient drop-outs. In my personal opinion (of course what I am going to say is not evidence-based either) a yearly frequency of visits is more realistic; patients should have a very easy way of contacting their clinicians, for example through telephone numbers that are answered directly by clinicians, so that in case of changes in the lesion patients can make an earlier appointment. I believe that this is associated with a lower rate of patient loss to follow-up, although, again, this comment of mine is not evidence-based either.

Response: Thank you for your statement. We know there isn´t enough evidence to justify our follow-up regime presently. Nevertheless, we have good experience with the short follow-up interval. Both in early detection of OSCC and in adherence/compliance of patients. Our median follow-up are 11years. By getting a regular medical referral from their dentist and coming to a consultation to a professor of oral medicine we believe our patients are in fact very informed about their OPMD and the risks going along with it. Even if they are not coming to every appointment, we can see small changes in their OLP sites do alert most of the patients. The solution we have in Germany (checkups twice a year covered by the insurance at their dentist) do not seem to lead to an early detection of OSCC in Germany compared to other countries. In our clinic we tried to make a difference. In our opinion it will be interesting to see if we are able to detect more OSCC in a very early stage. So far there are signs in our data leading to this conclusion.

Reviewer 2 Report

Comments and Suggestions for Authors

Early detection of malignant transformation of potentially malignant disorders: Oral Lichen Planus” by Ludeke at al deals with oral lichen planus. The value in the paper is that it follows a large number of oral lichen planus patients over some years to look for OSCC. There may be value to have something like a survival curve which marks detection of OSCC and not death to get a better idea what the average followup time was for the patients. Regardless I could not figure out what the average and median followup times were which may have been there in the manuscript but needs to be made more obvious. This is more like a case series then a research paper in that some controls are missing. Not clear how 3 month brush biopsy interval was derived.

It would also be nice to know OSCC incidence for patients that for example had benign leukoplakia that was not lichen planus related and how many of them eventually were diagnosed as OSCC and when after initial lesion detection. Even if brush biopsy was not used to monitor.

Line 37 “Lichen is an idiopathic immune-mediated inflammatory disorder with a prevalence…”

Should be Lichen Planus

Is not clear what the inclusion criteria are. It seems you are saying all patients with lichen planus that was surgically or brush biopsied that agreed to be in the study were included. And then must have a followup appointment 6 months or later and no prior history of Head and neck cancer? Is brush biopsy a valid diagnostic for OLP? In the abstract it suggests all patients had histopathogical diagnosis of oral lichen planus. Please make that clear. It is possible in some cases hisotpath. Not needed but more details need to be presented.

Line 131          Interestingly of 146 patients who had a referral, only 71 patients were diagnosed with oral lichen planus.     This is not very meaningful unless we are told why the patients were referred.

Line   Only 62 patients in our study showed a chance of clinical form during follow up.

Must be: “showed a change ….”

I am not sure how brush biopsy can be used to detect OLP. Need to hear criteria for judgement that person had oral lichen planus if they did not receive a surgical biopsy.  Without surgical biopsy and histopathology how do you know patients did not have Pemphigus for example. What criteria were used to rule out lichenoid response?

Need to cite Neummann et al. PMCID: PMC9643281  PMID: 35881238 and other papers that look at brush biopsy based detection of OSCC.

Comments on the Quality of English Language

Early detection of malignant transformation of potentially malignant disorders: Oral Lichen Planus” by Ludeke at al deals with oral lichen planus. The value in the paper is that it follows a large number of oral lichen planus patients over some years to look for OSCC. There may be value to have something like a survival curve which marks detection of OSCC and not death to get a better idea what the average followup time was for the patients. Regardless I could not figure out what the average and median followup times were which may have been there in the manuscript but needs to be made more obvious. This is more like a case series then a research paper in that some controls are missing. Not clear how derive 3 month interval for brush biopsy surveillance.

It would also be nice to know OSCC incidence for patients that for example had benign leukoplakia that was not lichen planus related and how many of them eventually were diagnosed as OSCC and when after initial lesion detection. Even if brush biopsy was not used to monitor.

Line 37 “Lichen is an idiopathic immune-mediated inflammatory disorder with a prevalence…”

Should be Lichen Planus

Is not clear what the inclusion criteria are. It seems you are saying all patients with lichen planus that was surgically or brush biopsied that agreed to be in the study were included. And then must have a followup appointment 6 months or later and no prior history of Head and neck cancer? Is brush biopsy a valid diagnostic for OLP? In the abstract it suggests all patients had histopathogical diagnosis of oral lichen planus. Please make that clear. It is possible in some cases histopath. Not needed but more details need to be presented.

Line 131          Interestingly of 146 patients who had a referral, only 71 patients were diagnosed with oral lichen planus.     This is not very meaningful unless we are told why the patients were referred.

Line   Only 62 patients in our study showed a chance of clinical form during follow up.

Must be: “showed a change ….”

I am not sure how brush biopsy can be used to detect OLP. Need to hear criteria for judgement that person had oral lichen planus if they did not receive a surgical biopsy.  Without surgical biopsy and histopathology how do you know patients did not have Pemphigus for example. What criteria were used to rule out lichenoid response?

Need to cite Neummann et al. PMCID: PMC9643281  PMID: 35881238 and other papers that look at brush biopsy based detection of OSCC.

Author Response

Early detection of malignant transformation of potentially malignant disorders: Oral Lichen Planus” by Ludeke at al deals with oral lichen planus. The value in the paper is that it follows a large number of oral lichen planus patients over some years to look for OSCC. There may be value to have something like a survival curve which marks detection of OSCC and not death to get a better idea what the average followup time was for the patients. Regardless I could not figure out what the average and median followup times were which may have been there in the manuscript but needs to be made more obvious. This is more like a case series then a research paper in that some controls are missing. Not clear how derive 3 month interval for brush biopsy surveillance.

Response: Thank you for your comment.

  1. Concerning survival curve: We created a Kaplan-Meier curve as you recommended. (figure 1)
  2. Concerning follow-up times: Thank you for pointing this out. We added the median follow-up. You can find it here: p.3, line 134: “The mean follow-up period was 6,02 years (6 months up to 22,2 years). The median follow-up time was 11,4 years.”

It would also be nice to know OSCC incidence for patients that for example had benign leukoplakia that was not lichen planus related and how many of them eventually were diagnosed as OSCC and when after initial lesion detection. Even if brush biopsy was not used to monitor.

Response: Thank you for your comment. In fact, we analyzed data of OLP patients only and can’t present the data on Leukoplakia patients yet.

Line 37 “Lichen is an idiopathic immune-mediated inflammatory disorder with a prevalence…”

Should be Lichen Planus

 Response: Thank you for the correction, we have updated the text.

Is not clear what the inclusion criteria are. It seems you are saying all patients with lichen planus that was surgically or brush biopsied that agreed to be in the study were included. And then must have a followup appointment 6 months or later and no prior history of Head and neck cancer? Is brush biopsy a valid diagnostic for OLP? In the abstract it suggests all patients had histopathogical diagnosis of oral lichen planus. Please make that clear. It is possible in some cases histopath. Not needed but more details need to be presented.

Response: Thank you for indicating this paragraph needs further clarification. All patents underwent clinical and histological diagnosis. Of course, OLP isn`t a valid tool for diagnosing OLP, it was used to control the biological behavior of lichen and to rule out malignancy.

“Oral lichenoid lesions were ruled out by removal of potential triggers (e.g. Amalgam) or excluded in this study. In cases of missing histopathological findings, a scalpel biopsy was performed to confirm the diagnosis of oral lichen planus. Moreover, a brush biopsy was performed to control the biological behaviour of cells”

“Follow up period lasted > 6 months from clinical & histopathological diagnosis.”

Line 131          Interestingly of 146 patients who had a referral, only 71 patients were diagnosed with oral lichen planus.     This is not very meaningful unless we are told why the patients were referred.

 Response: Thank you for bringing this up. We wanted to point out how many dentists seem to be insecure in diagnosing OPMDs correctly. We relate to it in the discussion (p.6, line 212). Therefore, we have made the following changes: p.3, line 137: “Interestingly of 146 patients who had a referral to our clinic by general dentists, only 71 patients were diagnosed with oral lichen planus, thus 48% of the cases were misdiagnosed.”

Line   Only 62 patients in our study showed a chance of clinical form during follow up.

Must be: “showed a change ….”

 Response: Thank you for pointing this out. We corrected the spelling error. (p.5, line 183: “change”)

I am not sure how brush biopsy can be used to detect OLP. Need to hear criteria for judgement that person had oral lichen planus if they did not receive a surgical biopsy.  Without surgical biopsy and histopathology how do you know patients did not have Pemphigus for example. What criteria were used to rule out lichenoid response?

Response:  Thank you for this comment. The brush biopsy was not performed to diagnose, but to rule out a potentially morphological suspicious cells/ an abnormal ploidy status. In follow-ups it ensured the normal ploidy status of the OLP lesion. We tried to clarify it by adding the statements:

p.3, line 95: “Oral lichenoid lesions were ruled out by removal of potential triggers (e.g. Amalgam) or excluded in this study. In cases of missing histopathological findings, a scalpel biopsy was performed to confirm the diagnosis of oral lichen planus. Moreover, a brush biopsy was performed to control the biological behaviour of cells.”

p.3, line 118: “Additionally, during follow-up brush biopsies were taken from clinically suspicious regions to enable a non-invasive, effective way of detecting a chance in ploidy status. The results were compared to the results of histopathological examination of the respective area.”

Need to cite Neummann et al. PMCID: PMC9643281  PMID: 35881238 and other papers that look at brush biopsy based detection of OSCC.

Response: Thank you for the recommendation. We added Neumann et al. and another paper (Bechstedt et al. doi:10.3390/cancers14235828.) to our citation list.

p.6, line 209: “Consistent with literature our study shows oral brush biopsies are an easy, inexpensive, reliable and effective tool to detect oSCC in early stages. [39] (p 6658)[29]”

Reviewer 3 Report

Comments and Suggestions for Authors

line 74: The text uses 'OSCC' and 'oSCC' – please use the same versions.

line 133-137: The number of cases of malignant transformation was too small, and it is recommended that the impact of sample size limitations on the research conclusions be emphasized in the discussion.

line 135/146: Please don't put references in the results. Add them to the discussion instead. 

line 156: What type of DNA image cytometry belonged for these four patients in Table 2? diploid (2c) or tetraploid (4c) cells or a single cell containing DNA >9c? 

line 136/161: Why is 97.13 months given instead of years?

line 167: "Oral Lichen planus" Pay attention to upper and lower case and keep the format consistent.

line 176-178: Clinical morphological changes are mentioned here, but it is not specified whether they are related to malignant transformation. Supplementary analysis or citation of supporting data is recommended.

Author Response

line 74: The text uses 'OSCC' and 'oSCC' – please use the same versions.

Response: Thank you for bringing this to our attention. Of course we fixed this inconsistency.

line 133-137: The number of cases of malignant transformation was too small, and it is recommended that the impact of sample size limitations on the research conclusions be emphasized in the discussion.

Response: We agree completely. To emphasize this point we stated: p.6, line 223:” Additionally, we had only 4 cases of malignant transformation, which is a very limited cohort and needs to be recognized.”

line 135/146: Please don't put references in the results. Add them to the discussion instead. 

Response: Thank you for your comment. We deleted all references in the results.

line 156: What type of DNA image cytometry belonged for these four patients in Table 2? diploid (2c) or tetraploid (4c) cells or a single cell containing DNA >9c? 

Response: Thank you for your comment. We added the precise results in Table 2.

line 136/161: Why is 97.13 months given instead of years?

Response: Thank you for the comment. We converted it to years. line 143 and 171: “8.1years”

line 167: "Oral Lichen planus" Pay attention to upper and lower case and keep the format consistent.

Response: Thank you for your mindful reading. We unified it to “oral lichen planus”

line 176-178: Clinical morphological changes are mentioned here, but it is not specified whether they are related to malignant transformation. Supplementary analysis or citation of supporting data is recommended.

Response: Thank you for your comment. We tried to clarify our statements and found supporting data. We hope this meets your expectations.

Line 183-192: “Only 62 patients in our study showed a change of clinical form during follow up. This could be due to the start and stop of treatment, the intake of triggers and the possibility of different clinical forms being present at the same time in one patient. In addition, the overall stability of the OLP lesions observed clinically in the present study seems to exclude the direct correlation between neoplastic events and changes in OLP lesions appearance over time [34] (p 331). However, changes in ploidy status or dysplasia are not visible, therefore periodic histologic or cytologic diagnostics are needed to detect these microscopic changes toward malignancy to eliminate guesswork about which lesion requires surgical biopsy and lessen the delay in referring patients [23].”

Round 2

Reviewer 2 Report

Comments and Suggestions for Authors

Early detection of malignant transformation of potentially malignant disorders: Oral Lichen Planus , by Ludecke et al. shows the results of a program that used frequent followup visits along with brush biopsy to monitor for transformation in Oral Lichen Planus. It is nice to get some data on Oral Lichen Planus transformation rates and to see that brush biopsy with chromatin staining may be a way to catch transfomation early. However the writers should avoid strong statement as they themselves cite references (line 195) that report lichen planus  linked cancers are often found at earlier stage even without brush biopsy. Second please do not include a survival curve. My question about how long it takes to transform from lichen planus, initial interaction with clinic, to OSCC is actually listed in Table 2 so the survival curve to look at OSCC incidence (not death) is not needed.   It is a shame no comparison of transformation rates for benign leukoplakia is done as that would certainly provide more insight on lichen planus and cancer risk.

Line 83

This fact leads to
reduced mortality and morbidity and a better survival rate in our clinic compared to the 84
literature.

Please change to: This may lead to reduced mortality and morbidity.... 

Without a control group for the population of lichen planus patients it is hard to know what is happening.

This may be in there and I missed it, but how many brush biopsy samples indicated aneuploidy yet were not dysplastic or OSCC by standard histopathology (False positives).

Author Response

Early detection of malignant transformation of potentially malignant disorders: Oral Lichen Planus , by Ludecke et al. shows the results of a program that used frequent followup visits along with brush biopsy to monitor for transformation in Oral Lichen Planus. It is nice to get some data on Oral Lichen Planus transformation rates and to see that brush biopsy with chromatin staining may be a way to catch transfomation early. However the writers should avoid strong statement as they themselves cite references (line 195) that report lichen planus  linked cancers are often found at earlier stage even without brush biopsy. Second please do not include a survival curve. My question about how long it takes to transform from lichen planus, initial interaction with clinic, to OSCC is actually listed in Table 2 so the survival curve to look at OSCC incidence (not death) is not needed.   It is a shame no comparison of transformation rates for benign leukoplakia is done as that would certainly provide more insight on lichen planus and cancer risk.

Response: Thank you for this comment. We removed the survival curve. As we stated before we can present data on OLP only. In future we will be able to compare behavior of OLP/ leukoplakia transformation.

Line 83

This fact leads to
reduced mortality and morbidity and a better survival rate in our clinic compared to the 84
literature.

Please change to: This may lead to reduced mortality and morbidity.... 

Without a control group for the population of lichen planus patients it is hard to know what is happening.

Response: Thank you for your comment. We changed it accordingly to your recommendation.

This may be in there and I missed it, but how many brush biopsy samples indicated aneuploidy yet were not dysplastic or OSCC by standard histopathology (False positives).

Response: As written in our article, there are 4 possible categories of brush biopsy. “negative”, “doubtful”, “suspicious” an “positive”. If the brush biopsy was something other than “negative” we performed a DNA-ICM to rule out any uncertainty regarding the ploidy status.

Therefore, we accept false positives in the first step (high sensitivity), because DNA-ICM will discover false positives afterwards. In the table below you can find the exact numbers.

Brush biopsy

no OSCC/false positive

Number of probes overall

positive

1

8

suspicious & doubtful

114

121

Round 3

Reviewer 2 Report

Comments and Suggestions for Authors

"Early detection of malignant transformation of potentially malignant disorders: Oral Lichen Planus" is an interesting paper on the usage of brush biopsy to monitor oral lichen planus. I find it acceptable for publication.